# Anti-Inflammatory Effects of Marine Bioactive Compounds and Their Potential as Functional Food Ingredients in the Prevention and Treatment of Neuroinflammatory Disorders

**DOI:** 10.3390/molecules28010002

**Published:** 2022-12-20

**Authors:** Mohamed Elbandy

**Affiliations:** Department of Clinical Nutrition, College of Applied Medical Science, Jazan University, Jazan 45142, Saudi Arabia; melbandy@jazanu.edu.sa; Tel.: +966-55-355-2971

**Keywords:** marine, functional food ingredients, neuroinflammation, neurodegenerative diseases, microglia, anti-inflammatory properties

## Abstract

Functional foods include enhanced, enriched, fortified, or whole foods that impart health benefits beyond their nutritional value, particularly when consumed as part of a varied diet on a regular basis at effective levels. Marine sources can serve as the sources of various healthy foods and numerous functional food ingredients with biological effects can be derived from these sources. Microalgae, macroalgae, crustaceans, fungi, bacteria fish, and fish by-products are the most common marine sources that can provide many potential functional food ingredients including phenolic compounds, proteins and peptides, and polysaccharides. Neuroinflammation is closely linked with the initiation and progression of various neurodegenerative diseases, including Alzheimer’s disease, Huntington’s disease, and Parkinson’s disease. Activation of astrocytes and microglia is a defense mechanism of the brain to counter damaged tissues and detrimental pathogens, wherein their chronic activation triggers neuroinflammation that can further exacerbate or induce neurodegeneration. Currently, available therapeutic agents only provide symptomatic relief from these disorders and no therapies are available to stop or slow down the advancement of neurodegeneration. Thereffore, natural compounds that can exert a protective effect against these disorders have therapeutic potential. Numerous chemical compounds, including bioactive peptides, fatty acids, pigments, alkaloids, and polysaccharides, have already been isolated from marine sources that show anti-inflammatory properties, which can be effective in the treatment and prevention of neuroinflammatory disorders. The anti-inflammatory potential of marine-derived compounds as functional food ingredients in the prevention and treatment of neurological disorders is covered in this review.

## 1. Introduction 

Inflammation is a complex biological reaction by which the human body responds to various harmful stimuli, including toxic compounds, damaged cells, and pathogens. In addition, inflammation plays a vital role in tissue repair and regeneration [1]. Generally, acute and controlled inflammation is beneficial, while chronic inflammatory response can occur due to inappropriate immune response and result in tissue injury and destruction [2]. As with inflammation, the term neuroinflammation denotes the process that occurs due to nervous system damage [3]. Chronic neuroinflammation can limit physiological control and result in various harmful actions, such as oxidative stress, pro-inflammatory signaling mechanisms, and neuronal death [4,5]. Multiple activated cells can play roles in neuroinflammation, including immune cells (such as infiltrating T cells and neutrophils as well as resident macrophages and mast cells), oligodendrocytes, astrocytes, and microglia in the central nervous system (CNS) and satellite glial cells and Schwann cells in the peripheral nervous system (PNS) [3,6]. Characteristics of neuroinflammation include glial cell activation or immune cell infiltration along with the generation of inflammatory mediators in the CNS and PNS [7]. Major neuroinflammatory disorders include pain, brain ischemia, epilepsy, Alzheimer’s disease (AD), Huntington’s disease (HD), and Parkinson’s disease (PD) [3,7,8,9,10,11,12,13,14]. Functional foods are enhanced, enriched, fortified, or whole foods that offer health benefits beyond basic nutrients (including minerals and vitamins) and energy by improving certain physiological responses and/or decreasing the risk of diseases [15]. On the other hand, functional food ingredients are bioactive compounds that have the potential to be utilized in the manufacturing of functional food products [16]. 

Nutrients and bioactive compounds derived from marine environments have greater potential as functional food ingredients since they can exhibit various beneficial effects along with medicinal properties and additional health benefits, including anti-inflammatory effects [17]. There are several sources from which marine-based bioactive food ingredients can be derived, such as marine sponges, microorganisms, and plants [18,19]. The effect of these naturally occurring bioactive substances in the human body might be insignificant over a comparatively short period. However, when consumed throughout life as part of the daily diet, these substances can play a significant role in improving health outcomes [20,21]. A number of chemical compounds that show anti-inflammatory properties, such as bioactive peptides, fatty acids, pigments, alkaloids, and polysaccharides, have already been isolated from marine sources, which could prove to be useful in the treatment and prevention of various neuroinflammatory disorders [22]. This review highlights various sources of functional food ingredients derived from a range of marine sources. Existing scientific knowledge demonstrating the role of inflammation in neuroinflammatory disorders has also been outlined in this review. Moreover, studies focusing on the anti-inflammatory potential of marine-derived compounds as functional food ingredients in the prevention and treatment of neurological disorders have been abridged in this review. This review seeks to provide new perspectives for research and industry regarding the emerging significance of marine bioactives as functional food ingredients in the prevention and treatment of a range of neuroinflammatory disorders 

## 2. Methods

The four most popular search engines, including PubMed, Google Scholar, and the *Web of Science and Scopus* databases, were utilized to obtain the relevant scientific literature. The terms and combinations of terms that were used to search the literature include marine functional food, marine bioactive compounds, neuroinflammatory disorders, and inflammation and neurological disorders. The terms marine and neuroinflammation have been combined with polysaccharides, astaxanthin, fucoxanthin, siphonaxanthin, proteins, peptides, amino acids, omega-3 fatty acids, and polyphenols. Furthermore, no restrictions to date and publication type were placed in the search of the literature. 

## 3. Inflammation in Neuroinflammatory Disorders

Chronic neuroinflammation can be harmful. Inflammation can play various roles in disease pathogenesis across the CNS (such as AD, multiple sclerosis (MS), PD, autism spectrum disorder, depression, traumatic brain injury, and motor neuron disease) and PNS (such as fibromyalgia and neuropathic pain) [23]. Multiple cells mediate neuroinflammation, including astrocytes, resident macrophages in the brain, and microglia (Table 1); however, these cells in non-pathological conditions make significant contributions to the physiology and metabolism of the brain and also play a significant role in maintaining the integrity of the blood–brain barrier (BBB) [24]. Acute neuroinflammation is likely to exert a protective effect in the body, while chronic neuroinflammation exerts harmful effects in the CNS [25,26]. Growing evidence indicates that the pathogenesis of neurological disorders such as MS, PD, and AD involves strong brain–immune interactions. A common characteristic of these disorders is the range of different interactions their neuropathological hallmarks have with astroglia and microglia. These interactions can induce an innate immune response and mediate the release of inflammatory mediators, which can further result in worsened severity and progression of diseases. 

In AD, Amyloid β (Aβ) is generated through the proteolytic processing of amyloid precursor protein (APP). This generated Aβ then forms aggregates that cause microglial activation via the receptor for advanced glycation endproducts (RAGE) and toll like receptor (TLR). Subsequently, these receptors cause activation of AP-1 and NF-κB transcription factors, which further triggers the generation of ROS and the expressions of various inflammatory cytokines, including TNF, IL-6, and IL-1 (Figure 1). Furthermore, these inflammatory factors can directly play roles in neurons and induce astrocytes, which further induces pro-inflammatory signals and eventually shows neurotoxic properties [46]. Necrosis and apoptosis of neurons can lead to further secretion of ATP, which can cause microglial activation via the purinergic P2X7 receptor. Microglia exerts protective functions by inducing Aβ clearance via apolipoprotein E (APOE)-dependent and -independent processes [47]. AD primarily affects the cholinergic neurons in the basal forebrain, and these neurons are supposed to serve as important targets of inflammation-caused toxicity. However, other types of neurons, such as GABAergic and glutaminergic neurons, are also affected by AD [47]. 

In the case of MS, viruses, bacteria-caused infections, or other environmental stimuli can induce the activation of astrocytes and microglia, which can result in the generation of pro-inflammatory cytokines via the activation of several transcription factors, including AP-1 and NF-κB (Figure 2) [47]. Naive T cells can detect myelin-derived antigens displayed in the context of MHC molecules in antigen-presenting cells [47]. In the occurrence of TGF-β and IL-6, naive T cells are stimulated to differentiate into Th17 cells and express retinoic acid receptor-related orphan receptor γt (RORγt). Activated astrocytes and microglia release osteopontin and IL-23, which can further stimulate Th17 cells to release IL-17. TNF-α can damage the myelin sheath that provides protection to nerve axons. Activated astrocytes generate B-cell activating factor (BAFF), which can differentiate into plasma cells and generate anti-myelin antibodies. Activated forms of astrocytes and microglia serve as the sources of nitric oxide (NO) and reactive oxygen species (ROS), which can eventually play a role in the obliteration of myelin sheaths and even neurons themselves. Regulatory T cells (Treg) that express transcriptional factor forkhead box P3 (Foxp3) inhibit the functions of Th17 cells and thus inhibit inflammation [47]. 

In PD, alpha-synuclein (α-Syn) is an important protein that is closely linked with pathogenesis. PD characteristics include the loss of dopaminergic neurons in the substantia nigra pars compacta and the aberrant aggregation and accumulation of α-Syn in the form of Lewy neurites and Lewy bodies (Figure 3). In PD, α-Syn aggregation is also linked with the degeneration and dysfunctionality of neurons [48]. In addition to the formation of Lewy bodies, α-Syn aggregates form intermediate-state oligomers that, when secreted from neurons, cause microglial activation via TLR-independent processes [47]. This can cause NF-κB activation and the generation of pro-inflammatory mediators and ROS. Furthermore, these factors can directly affect dopaminergic neurons of the substantia nigra, which predominantly die in the case of PD. These aforesaid factors can cause microglial activation, which induces inflammation in a positive feedback loop and causes further microglial activation. Astrocytes and microglia-derived products play combined roles in mediating neurotoxicity. Bacterial lipopolysaccharide (LPS) acts mainly via microglia-expressed TLR4, which can further trigger inflammation in the substantia nigra that leads to dopaminergic neuronal loss. Nuclear receptor related 1 (NURR1) is a ligand-activated transcription factor that inhibits inflammation in astrocytes and microglia by suppressing NF-κB target genes [47].

Neuroinflammation is facilitated via various ROS, secondary messengers (prostaglandins and NO), chemokines (CXCL1, CCL5, and CCL2), and pro-inflammatory cytokines (interleukin 6 (IL-6), tumor necrosis factor-alpha (TNFα), and interleukin-1β (IL-1β)). Most of these mediators are generated via activated resident immune cells such as astrocytes and microglia [49]. Furthermore, in the CNS, perivascular macrophages and endothelial cells play crucial roles in propagating and interpreting these inflammatory signaling pathways [50]. Cytokine generation and active microglia have also been detected in early brain development [51]. Active microglia also play a role by providing support, maintaining immunological functions, and performing synaptic pruning within the CNS [52]. Interestingly, IL-1 signaling pathways were found to play vital roles in the repopulation of depleted microglia from microglial progenitor cells [53]. Improved neuroinflammatory signaling pathways between resident CNS cells and T-cells have also been linked with normal learning and memory [54,55]. On the other hand, pathological or highly destructive neuroinflammation has been linked with CNS glia activation along with substantially elevated BBB breakdown and permeability, edema, infiltration of peripheral immune cells, and production of chemokines and cytokines [25,56,57]. 

In addition, there is a primary injury caused via physical and mechanical injury involving insult, damage, or infection. There can also be cell death, ischemia, vascular occlusion, and other secondary components of inflammation from these injuries. This extent of neuroinflammation has been linked with stroke, CNS infection, and various other diseases [58,59,60,61]. In most of the cases, these insults or injuries can be life-threatening, are linked to negative functional outcomes, and can mediate neuroinflammatory mechanisms with more pathological constituents. Moreover, this increased level of inflammation can result in both primary and secondary damage, which can further lead to chronic neuroinflammatory components that might never be resolved. Neuroinflammation has also been linked with immune responses mediated via autoimmune diseases such as MS, which has been exhibited in mouse models with experimental autoimmune encephalomyelitis (EAE) [60]. In this regard, the presence of auto-reactive T-cells, infiltration of peripheral immune cells, generation of chemokines and cytokines, and CNS glia activation have been observed [62]. A marked autoimmune response against myelin basic protein (MBP) has also been detected, which eventually resulted in the demyelination of axons. This inflammation can progressively become chronic and cause axonal loss [63]. Subsequently, this acute and chronic inflammation can affect myelin physiology, such as functional impairment mediated by axonal fragmentation and loss of myelin. Chronic neuroinflammation has already been linked with AD [61,64]. AD progression involves neuronal atrophy, neuronal injury and death, infiltration of peripheral immune cells, CNS glia activation, and protein misfolding [64,65]. Eventually, these chronic neuroinflammatory responses result in tissue injury. In AD and MS, the aforesaid tissue injury to neurons and axons has substantial functional outcomes. Moreover, neuroinflammation becomes progressive, chronic, and eventually destructive [66]. 

## 4. Functional Food Ingredients from Marine Sources 

### 4.1. Polysaccharides

Polysaccharides are polymers composed of chains of disaccharide or monosaccharide units. Polysaccharides are used widely in many commercial applications, including use as emulsifiers, thickeners, and stabilizers in beverages and foods [67]. Most of the polysaccharides that have been extensively studied as marine-derived functional ingredients are derived from microalgae and macroalgae sources (Table 2). The polysaccharide level in seaweeds can vary depending on the harvesting time, cultivation method, location, water temperature, season, geographical origin, and macroalgae species [17,67,68]. 

#### 4.1.1. *Fucans, Carrageenans*, and Fucoidans 

Fucans are sulfated polysaccharides that contain fucose backbones [81]. Former studies showed that they contain mainly (1→2) linked 4-O-sulfated fucopyranose residues. Nonetheless, subsequent studies have revealed that 3-linked fucose with 4-sulfated groups also exists in some of the fucose residues [82]. Multiple sulfated fucans, including carrageenans (from red algae), fucoidans (from brown algae), and ulvans (from green algae), show various important effects, such as coagulation modulation, anticancer, antidiabetic, antioxidant, antithrombotic, and anti-inflammatory properties [83,84]. On the other hand, soluble polysaccharides derived from algae are being assessed as possible new prebiotic compounds and possess great potential as dietary fiber for human nutrition [85]. 

Seaweeds serve as the main source of sulfated polysaccharides and have multiple biological functions. It has been observed that the structures of these molecules can vary in relation to the species of algae [86]. The major type of sulfated polysaccharide present in red algae is galactan, which is composed completely of galactose or modified galactose units. Brown algae mainly contains fucans that are composed of a group of sulfated L-fucose-based polydisperse molecules. Most of the water-soluble polysaccharides present in green seaweed are ulvans [82], and some of them serve as a major source of sulfated galactans in *Codium* species (Chlorophyta, green algae) [87,88]. It has been demonstrated that sulfated polysaccharides exert numerous biological effects, including anti-inflammatory, antioxidant, antiadhesive, antiproliferative, antiviral, antitumor, and anticoagulant properties [89]. In the food industry, sulfated galactans are also extensively utilized because of their thickening and jellifying properties [90]. Some red seaweeds contain various other sulfated polysaccharides, such as xylomannan sulfate and carrageenan. The carrageenan hydrocolloid is a linear sulfated polysaccharide of 3,6-anhydro-Dgalactose and D-galactose that is obtained from some red seaweeds. There are three basic types, specifically lambda carrageenans, kappa carrageenans, and iota carrageenans; however, seaweeds generally do not generate pure carrageenans, instead they produce various hybrid structures [91,92]. 

Copolymers of iota- and kappa-carrageenan jellify with distinct characteristics and are utilized in food production because of their outstanding functional and physical characteristics (such as stabilizing, jelling, and thickening properties); these copolymers can be found in many products, including jams, ice cream, puddings, and yogurt. Moreover, they are used in multiple non-food industries, such as in pharmaceutical products, owing to their anti-inflammatory and anticoagulant properties [93]. Fucoidans are a group of sulfated polysaccharides that possess sulfate ester groups and α-1,3-linked sulfated L-fucose as its key sugar unit [68]. It has been observed that the number of sulfate groups affects the antiangiogenic and antitumor properties of fucoidans [94]. Nonetheless, fucoidans exert various other important biological functions, including anti-inflammatory, antitumor, antioxidant, and anticoagulant effects [89,94,95,96]. Porphyran is a sulfated polysaccharide derived from red seaweeds and shows various properties, including antitumor, antioxidant, and immunoregulatory properties [97]. 

#### 4.1.2. Agar and Laminarin

Agar is a water-soluble long-chain polysaccharide extracted from agarophyte red algae, which is extensively utilized in the gelatin manufacturing process. Agar is composed of two major components: agaropectin and agarose. Agaropectin shows low gelling properties [67]. On the other hand, agarose is a linear polymer that shows a higher level of gel-forming properties. Agarose is composed of a disaccharide repeating unit consisting of 4-linked 3,6-anhydro-1-galactose and 3-D-galactose residues, with probable sulphate, methoxy, and other substituents in the polysaccharide chain [98]. It has been demonstrated that agar has the ability to exert an anti-aggregation effect on red blood cells and also reduce blood glucose levels. Laminarin is a common polysaccharide present in marine brown algae. This polysaccharide has the ability to modulate intestinal metabolism via its functions on short-chain fatty acid generation, mucus composition, and intestinal pH [99]. The antioxidant property of laminarin varies depending on its chemical structure and molecular weight. The chemical structure is composed of (1→3)-β-d-linear glucan along with some (1→6)-linkages [100]. The level of laminarin in seaweeds is around 10% of the dry weight; however, this level can increase to up to 32% depending on the season [67]. 

#### 4.1.3. Alginate

Alginates are linear unbranched polysaccharides that can be isolated from brown seaweeds, wherein they can comprise up to 47% of dry biomass. Alginate is a type of natural polysaccharide that shows excellent immunogenicity and biodegradability. Alginates have the capacity to form a gel with multiple cross-linking agents; this gel is used in various fields, including biotechnology, medicine, and the food industry [101]. Alginic acid is an algal polysaccharide that shows antiviral, anticoagulant, and antitumor properties. In rats, alginic acid was shown to have the ability to avert obesity, diabetes, and cancer of the large intestine, as well as decrease low-density lipoprotein [102,103,104,105]. Along with the aforesaid effects, it has already been demonstrated that alginic acid derivatives and fucoidans derived from brown seaweeds (including *Undaria pinnatifida*, *Ascophylum nodusum*, and *Ecklonia cava*) can play the role of antioxidants and free radical scavengers in the prevention of oxidative stress [97]. The anticoagulant and antioxidant properties of polysaccharides are greatly reliant on molecular weight, the molar ratio of sulfate and fucose/sulfate/total sugar, and the level of sulfate groups [95]. 

#### 4.1.4. Chitin and Chitosan Derivatives

Chitin (another marine polysaccharide) is widely found in shellfish waste and crustaceous shells. Crayfish, prawns, lobster, crab, and shrimp contain 14–35% chitin [17]. Chitosan, chitin, and derivatives including chitooligosaccharides can potentially be used as functional ingredients as they might elevate dietary fiber levels and reduce lipid absorption. Chitin–glucan complex (CGC) is a copolymer that has been approved as a fiber-boosting weight management product by the European Union (EU) [106]. Chito-oligosaccharides show various biological effects, including anticoagulant, hypocholesterolemic, hypoglycemic, antioxidant, anticancer, angiotensin-I-converting enzyme inhibition, and antimicrobial properties [107].

#### 4.1.5. Exopolysaccharides 

Exopolysaccharides (EPS) generated by extremophilic deep-sea bacteria and cyanobacteria have the potential to be used as functional food ingredients. Interestingly, the harsh environment of the ocean might trigger marine microorganisms to generate distinctive EPS [108]. Most of the EPS that are derived from marine microorganisms are heteropolysaccharides that consist of several monosaccharides (such as pyruvate, glucuronic acid, galactose, and glucose) in a certain ratio [98,108,109,110,111]. Polysaccharides generated via *Cyanospira capsulate* strains possess excellent viscosity properties that are similar to those of xantham gums, while EPS generated via *Vibrio* (bacteria), *Alteromonas*, and *Pseudoalteromonas* can serve as effective thickening agents [112]. Various complex polysaccharides derived from *Chlorella pyrenoidosa* (green algae, Chlorophyta) generates complex polysaccharides, including β-1,3-glucan with potential antioxidant and immunostimulating properties [17,113]. 

### 4.2. Pigments

Marine red algae, including *A. platensis* and *Porphyridium cruentum*, can generate up to 8% of phycobiliproteins, which are used for fluorescence studies when covalently attached to hormones, lectins, biotin, A-protein, and antibodies. Porphyridium’s phycoerythrin and Arthrospira’s phycocyanin are the best-known phycobiliproteins. These phycobiliproteins can be utilized as non-radioactive markers in DNA assays, microscopy assays, and fluorescence-based immunology assays [114]. Carotenoids impart red, orange, and yellow hues to exoskeletons, shells, and skin in various aquatic species. Numerous types of carotenoids are found in algae. Various carotenoids (particularly β-carotene) can play an important role as precursors of vitamin A in mammals. Vitamin A activity has also been confirmed in echinenone, ethyl ester of β-apo-8’-carotenoic acid, 3,4-Didehydro-beta-carotene, canthaxanthin, cryptoxanthin, and α-carotene. Marine carotenoids were found to suppress lipase function and decrease triacylglycerol absorption. Fucoxanthin (a marine carotenoid) is widely found in brown seaweeds [115,116], while astaxanthin is generated by various marine microorganisms, such as yeast (including *Phaffia rhodozyma*) and algae (including *Haematococcus*, *Chlorococcum*, and *Chlorella zofingiensis*) [117].

### 4.3. Proteins, Peptides, and Amino Acids

Proteins contribute as transport (for example hemoglobin), antibodies, and hormones (for example insulin) in food and biological systems. Seaweeds contain numerous nutrients and have various medicinal properties. Seaweeds have been utilized as food or herbal medicine for a range of ailments and diseases [118]. Moreover, seaweeds are used as fungicides, animal feed, herbicides, dietary supplements, and carrageenans in many pharmaceutical and industrial applications. Seaweeds have been serving as a source of protein for several decades [119]. 

There is a growing research interest regarding the hydrolysis of proteins for bioactive peptide synthesis, particularly peptides derived from marine proteins. Various peptides have already been extracted from marine fish, algae, and crustaceans. Bioactive peptides often contain in between 3 and 20 amino acids depending on the amino acid sequence and content [118]. These peptides exhibit various properties, including antibacterial, anticancer, immunomodulatory, antihypertensive, antithrombotic, and antioxidant properties [120]. Peptides extracted from marine protein sources might serve as important antihypertensive compounds and have the potential to be incorporated into functional foods [118,121,122].

Taurine is an abundant amino acid found in seafood muscles. Since water-soluble components are lost during thermal processing or cooking, their beneficial properties are likely to be retained when seafood is lightly prepared. It has been observed that the diets of children (particularly when they are based on grains) often have a deficiency of various sulfur-containing amino acids, including threonine and lysine. Therefore, the consumption of fish muscle proteins that are rich in certain amino acids might ameliorate human nutrition by increasing the nutritional value of foods [118,123,124].

### 4.4. Omega-3 Fatty Acids

Eicosapentaenoic acid (EPA) and docosahexaenoic acid (DHA) are the common omega-3 polyunsaturated fatty acids (PUFAs) produced from aquatic sources. Linolenic acid in humans can be elongated and desaturated to EPA and, to a lesser extent, DHA. In addition, these PUFAs could be obtained via supplementation and seafood. Compared to DHA and EPA, a lower level of docosapentaenoic acid (DPA) is seen in fish oils. Nonetheless, DPA is nearly as important as DHA or EPA in seal blubber oil. Multicellular and unicellular marine plants, including algae and phytoplankton, can generate DHA and EPA. Furthermore, they enter the food chain and eventually end up in the lipids of aquatic animals such as marine mammals and fish. Various in vitro and animal studies have shown that omega-3 fatty acids have the capacity to alter cell signaling pathways, blood lipid profiles, eicosanoid generation, cardiovascular health, membrane lipid composition, and gene expression. Intake of fatty acids from natural sources might affect the progression and initiation of many disease conditions, such as cancer, autoimmune, and cardiovascular diseases [125]. Along with PUFAs, an increased level of monounsaturated fatty acids (MUFAs) is also present in seafood. Both PUFAs and MUFAs are regarded as healthy if not oxidized [126].

### 4.5. Polyphenols

Phenolic compounds or polyphenols are present at increased concentrations in algae. These compounds show potent antioxidant, antimutagenic, and anti-inflammatory properties. Algae polyphenols are linked to chemical defenses against various external conditions, including herbivores and stress [127]. Multiple seaweed species are rich in polyphenolic compounds, such as *Porphyra* sp. (red algae, Rhodophyta), *Fucus* sp. (brown algae, Phaeophyceae), *Laminaria* sp., and *Undaria* sp. [128]. Phlorotannins, flavonols, and catechins have already been studied as functional food ingredients [129,130]. Phlorotannins have already been detected in various brown algae. It has been observed that these compounds are highly hydrophilic in nature and serve as inhibitors of metalloproteinase and hyaluronidase, while also showing anti-allergic, antihypertensive, antitumor, antidiabetic, anti-inflammatory, and antioxidant properties [86,127,131].

## 5. Potential of Marine-Derived Compounds as Functional Food Ingredients against Neuroinflammatory Disorders

A range of marine-derived compounds have already exhibited their potential as functional food ingredients with additional health benefits that include anti-neuroinflammatory properties. In Table 3, the potential marine-derived compounds that could be used as functional food ingredients against various neuroinflammatory disorders are outlined. 

### 5.1. Polysaccharides 

Neuroinflammation is responsible for the initiation and progression of multiple neurodegenerative disorders, including AD and PD [147]. Astrocytes and microglia provide protection for the brain against infectious agents; however, prolonged activation of microglia can result in neuroinflammation that can mediate neurodegeneration [147]. No effective therapy is available to stop the advancement of neurodegeneration. Since neuroinflammation plays a significant role in the initiation and advancement of neurodegenerative pathogenesis, anti-inflammatory molecules might therefore be good candidates for the development of effective therapeutic approaches [147,148].

Natural polysaccharides extracted from natural sources possess multiple monosaccharide units that are linked with each other via various glycoside linkages with a multifaceted molecular arrangement [149]. In addition, polysaccharides have an important role in pharmaceutical applications owing to their potent immunomodulatory and anti-inflammatory effects [149]. Seaweeds serve as a source of potential bioactive polysaccharides. Polysaccharides derived from brown seaweeds exhibit numerous bioactive properties. Fucoidans are sulfated polysaccharides derived from brown seaweeds. The structures of fucoidans vary across the seaweed species [150]. Fucoidans (Figure 4) exhibit a range of biological effects, including antioxidant, neuroprotective, and anti-inflammatory properties [151]. 

It is well known that activation of microglia can lead to the generation of intracellular and extracellular OS. Intracellular ROS are important for microglial survival and pro-inflammatory activity [41]. ROS generation is considered as the first step in cytokine generation in microglia. In addition, intracellular ROS can serve as the second messengers controlling various downstream signaling molecules, such as NF-κB, MAPKs, and protein kinase C, thus inducing the pro-inflammatory activity of microglia [152]. Various studies have reported that fucoidan exhibits anti-inflammatory effects by inhibiting extracellular ERK and p38 phosphorylation in primary microglia [153,154]. Collectively, these findings suggest that fucoidan has the capacity to decrease the generation of LPS-stimulated ROS [132]. Furthermore, this decreased level of ROS generation may be an important mediator of the neuroprotective role of fucoidan toward LPS-caused damage, such as the downregulation of MAPK signaling pathways and suppressive actions on NO generation and TNF-α secretion. In PD, microglia have a significant contribution to the initiation of a self-propagating cycle that can result in prolonged chronic neuroinflammation and also regulate progressive neurodegeneration. In a previous study, Cui et al. [132] reported that fucoidan can suppress LPS-induced microglial activation, protect dopaminergic neurons, and attenuate neuroinflammation. 

Acute intranigral injection of LPS resulted in rapid microglial activation and the subsequent generation within 24 h of various neurotoxic pro-inflammatory factors, including IL-1β, TNF-α, and ROS [155], which may further initiate the pathways that result in the death of neighboring dopaminergic neurons [156]. Cui et al. [132] also evaluated the activities of fucoidan on the activation of microglia and the generation of pro-inflammatory factors in primary microglia. TNF-α is a major pro-inflammatory cytokine, and overproduction of TNF-α may be closely linked to the death of dopaminergic neurons. It was observed that fucoidan treatment inhibited the increased level of TNF-α mRNA expression and TNF-α secretion mediated by LPS, which provides evidence of the neuroprotective activity of fucoidan in LPS-treated animals. In a previous study, Jhamandas et al. [157] revealed that fucoidan provides protection against Aβ-caused neurotoxicity in basal forebrain neuronal cultures. In a 1-methyl-4-phenyl-1,2,3,6-tetrahydropyridine (MPTP) animal model of PD, fucoidan also exerted neuroprotective properties on dopaminergic neurons [158]. Numerous studies have suggested that fucoidan has the capacity to control the secretion of pro-inflammatory cytokines from microglial cell lines. In another study, Do et al. [153] showed that fucoidan can inhibit iNOS expression and NO generation in IFN-γ- and TNF-α-induced C6 glioma cells. Park et al. [159] also showed that treatment with fucoidan can markedly suppress the overproduction of PGE_2_ and NO, as well as attenuate the expressions of IL-1β, monocyte chemoattractant protein-1 (MCP-1), TNF-α, and COX-2 in LPS-induced BV2 microglia [132]. In addition, treatment with fucoidan extracts ameliorated signs of neurodegeneration, learning, and memory in mouse models [160]. Interestingly, sulfated polysaccharides derived from the red alga *Gelidium pristoides* incubated with Aβ1–42 cleared Aβ1–42 fibrils, which further indicates the disaggregation and suppression role of Aβ1–42 fibrils [161]. 

Carrageenans are natural polysaccharides derived from certain species of red seaweeds. Carrageenan oligosaccharides show various physiological functions and exceptional properties, including increased absorption efficiency, higher water solubility, and low molecular weight. It has been observed that the biological functions of carrageenan oligosaccharides are closely linked with their structures, particularly the position and number of sulfate groups [162]. Moreover, kappa-carrageenan oligosaccharides (KOS) (Figure 5) exerted immunomodulatory activity by influencing LPS-activated microglial cells, which can be useful in averting inflammation-related neurodegenerative disorders [133]. It has been observed that LPS can induce TLR4 expression on the cell membrane of microglia [163]. In a previous study, Yao et al. [133] indicated that a competitive relationship might exist between KOS and LPS. There might be KOS-specific binding proteins on the surface of microglia, which may mediate KOS in the cell. LPS molecules containing a negative charge have the capacity to combine with microglia and trigger them into generating various inflammatory cytokines [133]. Interestingly, pretreatment with KOS or KOS derivatives prevented microglia from binding with LPS [133]. However, the protective function of KOS derivatives was comparatively weaker owing to their lower level of negative charge. Collectively, these findings suggest that the use of KOS might be useful in preventing the processes of CNS diseases. However, further research is required to explore the potential of KOS in the treatment of inflammation-associated neurodegenerative diseases [133].

### 5.2. Pigments

Carotenoids are the most common types of pigments found in marine environments. In general, these pigments are biosynthesized by all autotrophic marine organisms, including fungi, algae, archaea, and bacteria [70,164]. 

#### 5.2.1. Astaxanthin 

Among carotenoids derived from various marine organisms, astaxanthin (the red pigment) (Figure 6) shows potent scavenging properties against free radicals and various other pro-oxidant molecules by providing protection for lipid bilayers against peroxidation [70]. It has been observed that these effects are linked to their unique molecular structures; thus, astaxanthin shows a 10-fold higher antioxidant effect compared to other carotenoids, including β-carotene, canthaxanthin, and lutein [165]. Astaxanthin was found to increase the anti-apoptotic index ratio of Bax/Bcl-2. In addition, astaxanthin inhibited neuroinflammation by reducing the expression of COX-2 [166]. The effect of astaxanthin on neurodegeneration was linked to a decrease in histopathological alterations in the hippocampus, and an elevated level of BDNF expressions was observed in the hippocampi and brains of aging rat models [167]. In another study, Aslankoc et al. [168] observed that treatment with astaxanthin at a dose of 100 mg/kg for one week provided protection against methotrexate-induced damage in the cerebellar cortex, cerebral cortex, and hippocampus in rat models. In contrast, in the control group, there was an elevated level of oxidative damage in the blood, hippocampus, and cerebral cortex, as demonstrated by a reduction in total antioxidant status and an elevated total oxidative state, while astaxanthin decreased oxidative damage in the study group [135]. 

Astaxanthin also inhibited histopathological alterations in the cerebellar cortex, cerebral cortex, and hippocampus, including degenerative changes, edema, and congestion in the control groups. It has been confirmed that astaxanthin’s anti-inflammatory and anti-apoptotic effects are related to elevated MBP expression and decreased levels of caspase-3, iNOS, granulocyte colony-stimulating factor, and growth-related oncogene [168]. In another study, Zhao et al. [137] indicated that astaxanthin can be useful in the treatment of neuropathic pain. Astaxanthin was administered at a dose of 5–10 mg/kg from the fifth postoperative day in C57BL/6 mice for 23 days. In addition, astaxanthin partially reduced neuropathic pain and exhibited analgesic activity on day 7. Moreover, astaxanthin decreased microglial activation and the expression of various pro-inflammatory cytokines, which further resulted in the suppression of neuroinflammation. The anti-inflammatory activity of astaxanthin was demonstrated by the nuclear translocation of NFκB p65, as well as by the suppression of p38 and Erk1/2 phosphorylation [135,137].

#### 5.2.2. Fucoxanthin 

It is estimated that fucoxanthin (an orange-colored pigment) contributes over 10% of the total production of carotenoids in nature, particularly in the marine environment [169,170]. Fucoxanthin exerts a range of biological effects owing to its unique molecular structure. It contains an unusual allenic bond and various oxygen-containing functional groups, including the carboxyl, carbonyl, hydroxyl, and epoxy moieties in its molecule that play a role in its unique structure (Figure 6) [171]. Fucoxanthin is a promising carotenoid that has the potential to be utilized in CNS disorders. A major source of fucoxanthin is brown algae, including *Cladosiphon okamuranus, Laminaria japonica, Alaria crassifolia*, *Hijikia fusiformis*, *Undaria pinnatifida*, and *Sargassum siliquastrum* [70]. The biological effect of fucoxanthin was found to be linked to its potent anti-inflammatory effect [172,173]. It has been revealed by in vitro studies that fucoxanthin reduced LPS-caused neuroinflammation by reducing the secretion of various inflammatory mediators, such as IL-6, PGE-2, IL-1β, NO, and TNF-α [136]. Various other processes were also found to be linked to the anti-inflammatory potential of fucoxanthin, including reduced expression of iNOS and COX-2 and suppression of Akt/NF-κB and MAPK/AP-1 signaling cascades [136]. Fucoxanthin also inhibited neuroinflammation by regulating NLRP3 inflammasome. It has been demonstrated that fucoxanthin might regulate the initiation step of inflammasome signaling cascades since it has the potential to decrease the expression of phosphorylated IκBα and pro-IL-1β [135,173]. 

#### 5.2.3. Siphonaxanthin

Siphonaxanthin (Figure 6) is a keto-carotenoid found in various edible green algae, including *Umbraulva japonica*, *Caulerpa lentillifera*, *and Codium fragile* [135]. In the transfected human monocytic cell line, it was observed that treatment with siphonaxanthin at a concentration of 1.0 µM for 24 hours exhibited an anti-inflammatory effect, as demonstrated by the marked suppression of TNF-α- and LPS-induced NF-κB activation. Furthermore, siphonaxanthin pretreatment at a concentration of 1.0 µM markedly inhibited the IL-1β-mediated activation of NF-κB [134,135].

### 5.3. Proteins, Peptides, and Amino Acids

Bioactive peptides are regarded as important functional food ingredients. Many marine species serve as rich sources of bioactive peptides [122]. Bioactive peptides were first isolated and discovered in various marine species as antimicrobial peptides, cardiotoxin, antitumor peptides, antiviral peptides, cardiotonic peptides, and neurotoxins [122]. Since then, they have been extensively studied to reveal their applications. Interestingly, multiple bioactive peptides with anti-inflammatory potential have already been isolated from marine microalgae, bacterium, and sponges [174]. The bioactive peptide derived from *Pyropia yezoensis* (an important marine algae), PPY1, was evaluated for its anti-inflammatory potential. *Pyropia yezoensis* (*P. yezoensis*) is a member of a diverse group of photosynthetic marine organisms. Furthermore, it possesses the ability to adapt to survive in extremely competitive and complex environments, such as environments with low and high tides, low light intensities, temperature variations, and increased levels of salinity [175]. Thus, certain peptides isolated from *P. yezoensis* have the potential to exert important biological properties, including anti-inflammatory properties. 

It has been confirmed that the *P. yezoensis*-derived peptides have effects on cell proliferation and related signaling cascades in MCF-7 and IEC-6 cells [176,177,178,179]. In general, short peptides containing 2-10 amino acids exhibit more bioactive properties than large polypeptides or their parent native proteins. Five amino acids containing PPY1 were isolated from *P. yezoensisa* through enzymatic hydrolysis [140]. It was observed that PPY1 exerts anti-inflammatory effects by inhibiting inflammatory cytokines. This finding was consistent with the observation that the activity of any peptide is greatly reliant on its composition of amino acids [180]. In addition, amino acids containing Lys, Ala, and His were found to exert potent effects against the intracellular generation of ROS and NO [181]. In LPS-stimulated RAW 264.7 macrophages, it was reported that PPY1 might play a role in averting inflammation by suppressing the expressions of various inflammatory mediators, including COX-2 and iNOS, without having any activity on cell viability [140]. 

IL-1β and TNF-α are well-known inflammatory cytokines, and their levels are increased in the pathogenesis of many inflammatory and infectious diseases [182]. On the other hand, MAPKs control multiple immune and inflammatory responses, such as LPS-mediated expressions of iNOS and COX-2 in macrophages. Suppression of various MAPK family members, including JNK (c-Jun N-terminal kinase), p38, and ERK, resulted in the generation of various pro-inflammatory cytokines being blocked, which further led to the inhibited phosphorylation of these MAPK members. More studies are required to assess the effect of PPY1 in the treatment and prevention of various inflammatory-related disorders, including aging and neurodegenerative disorders [140]. 

Lectins are carbohydrate-binding proteins that selectively recognize carbohydrates and reversibly bind with them [183]. Lectins have a deficient enzymatic function on their ligand and are different from antibodies and free oligosaccharide and monosaccharide transport/sensor proteins [184,185]. Marine algae are good sources of lectins, particularly Phylum Rhodophyta or red algae. The marine red alga *Solieria filiformis* (Kützing) P.W. Gabrielson is extensively found along the northeast coast of Brazil. Its lectin (SfL) is monomeric in nature and shows affinity toward the carbohydrate mannan [139,186]. Treatment with lectin at doses of 1, 3, and 9 mg/kg showed anti-inflammatory properties in acute inflammation experiments in rat models and anti-depressive activity in animal models of depression [138,139]. Moreover, SfL at this dose did not alter the patterns of animal locomotion in open-field tests. Administration of SfL for one week did not exhibit visible signs of toxicity in mice [139,187]. 

### 5.4. Omega-3 Fatty Acids

Essential fatty acids (EFAs) are a type of PUFA that needs to be obtained dietarily, since they cannot be synthesized in the body [188]. PUFAs are divided into two families: omega-3 (ω-3) and omega-6 (ω-6) [189]. Compared to ω-6 FAs, the consumption of ω-3 is generally inadequate owing to limited sources. Typical sources of α-linolenic acid include canola oils, soybean, walnuts, flaxseed, and green leafy vegetables. Their derivatives, including eicosapentaenoic acid (EPA) (Figure 6) and docosahexaenoic acid (DHA) (Figure 7), are found in algae and fish oils from rainbow trout, herring, anchovies, sardines, mackerel, and salmon [190,191,192,193]. The major sources of long chain (≥C_20_) *n-3* PUFAs are seafoods such as molluscs (including squid and oysters), crustaceans (including lobsters), and fish [194]. DHA and EPA are important for human physiology. These omega-3 fatty acids play important roles in various brain functions, including synaptic plasticity, enzymatic function, and cellular metabolism [195]. Both EPA and DHA significantly contribute to the brain and the development of nerve cells [196].

It has been reported that diets rich in n-6 fatty acids and saturated fat can elevate the risk of developing neurological disorders, have negative effects on cognitive functions, and elevate insulin resistance [197,198,199,200]. Interestingly, intake of DHA and EPA was found to be linked to regulation of the neural membrane’s permeability and fluidity. Moreover, their consumption ameliorated spatial learning via the control of cognitive and synaptic functions [201,202]. Along with the intake of phytochemicals (including resveratrol and other polyphenols), increased consumption of whole grains, fish, vegetables, and fruits was found to be beneficial for the human brain [203,204,205]. It is well known that the Western diet contains increased levels of arachidonic acid (ARA), with the ratio of ARA to DHA being around 20:1. This ratio was 21:1 in the paleolithic diet. The paleolithic diet was also rich in fish, lean meat, vegetables, and fruits [206,207,208]. Increased consumption of ARA-rich food in the modern Western diet increased the concentration of thromboxanes, leukotrienes, and prostaglandins, as well as upregulated the expression of multiple pro-inflammatory genes, including genes for cytokines (IL-1β and TNF-α) and enzymes (NOS, COX-2, and secretory phospholipase A2). It has been observed that the aforesaid enzymes and genes begin and maintain neuroinflammation. On the other hand, intake of diets rich in DHA and EPA mediates anti-inflammatory activities that are partially mediated via the suppression of genes that encode pro-inflammatory cytokines [141]. As the Western diet is rich in ARA and low in DHA, this diet is thus associated with numerous chronic visceral diseases, as well as neuropsychiatric and neurodegenerative disorders [202,206,207,208].

It has already been demonstrated that consumption of DHA and EPA is related to a decreased risk of age-associated cognitive deficit [202,209]. In the brain, DHA is greatly concentrated and improves synaptic functions in neurons. Moreover, DHA decreases neuronal death following ischemic injury by controlling the biophysical properties of the neural membrane and by preserving activities in the pre- and post-synaptic regions. Indeed, in ischemic injury, these mechanisms led to better maintenance of intracellular ion balance. In the brain, DHA and EPA also averted apoptotic cell death via the production of neuroprotectins and resolvins, as well as by showing antiapoptotic properties such as the maintenance of mitochondrial function and integrity, downregulated expression of apoptotic proteins, upregulated expression of antiapoptotic proteins, and decreased response to ROS [202,210]. A high level of DHA is found in the membrane structures found at synaptic terminals. Low tissue concentrations of DHA and EPA were found to be linked to an elevated risk of AD development, memory loss, schizophrenia, and depression [202]. Low dietary consumption of DHA and EPA was also found to be linked to the cognitive deficit in individuals with AD.

Collectively, the aforesaid findings suggest that dietary enrichment with DHA and EPA might ameliorate oxidative damage and inflammation in neurodegenerative and neurotraumatic diseases through the actions of DHA and EPA on the physicochemical characteristics of neural cell membranes, as well as through the regulation of genes and the production of NPD1 and D-series resolvins [141,202]. It has been reported that DHA can elevate the anti-inflammatory phenotype of microglia by altering the composition of the cell membrane and by affecting the production of lipid mediators [142]. In addition to this, omega-3 fatty acids can elevate the phagocytosis of myelin debris and also mediate extracellular Aβ clearance from the brain in the case of AD, which is well known as the major cause of neuroinflammation [143,211,212,213]. In the AD brain, EPA and DHA levels were found to dramatically decrease with age [214]. EPA and DHA play a significant role in inhibiting inflammatory cytokines and controlling inflammatory responses. DHA treatment decreased M1 markers (including CD86 and CD40) and increased various phagocytic markers (including CD163 and CD206) in CHME3 cells [143,213,215]. These findings directly indicate the relationship between the phagocytic phenotype of microglia and DHA [143]. IL-6 generation was found to be decreased in microglia in a concentration-dependent manner. Collectively, these findings indicate the anti-inflammatory potential of omega-3 fatty acids [142,143,144,145].

Nevertheless, previous findings have indicated that short-term intake of dietary DHA alone might not stop cognitive deficit in AD; however, long-term DHA and EPA supplementation from childhood to old age might reinstate the signal transduction mechanisms associated with learning activity and behavioral impairments, and may also generate various immunological and neuroendocrinological effects on brain tissues that may further result in beneficial effects in AD and ischemia [141]. Therefore, more studies, particularly clinical trials, are essential to evaluate the efficacy of DHA and EPA in AD treatment.

### 5.5. Polyphenols

Microglial overactivation and the neuroinflammation that follows cause synaptic dysfunction and loss. Therefore, processes to control the activation of microglia might decrease neuronal death or injury in the case of neurodegenerative disorders. There is a growing interest in the anti-neuroinflammatory effect of phlorotannins [146,216,217,218,219]. Furthermore, in vitro studies have revealed the effects of phlorotannins at different important stages of LPS-induced inflammation [216,217,218,219,220]. Dieckol (Figure 8) is an important marine polyphenol and a phlorotannin isolated from brown seaweed [221]. Dieckol has the ability to act at the transcriptional level. This marine polyphenol significantly reduced LPS-mediated cytokine generation by inhibiting the expression of COX-2 and iNOS [216]. It was observed that dieckol exerts anti-neuroinflammatory effects by exerting antioxidant properties in BV2 microglia and by blocking the activation of p38 MAPK and NF-κB [216]. Dieckol also inhibited the microglia-mediated neurotoxicity observed in the pathogenesis of neurodegeneration and neuroinflammation by inhibiting the activation of microglia. This activity was found to be facilitated by the downregulation of nicotinamide adenine dinucleotide phosphate (NADPH) oxidase, extracellular signal-regulated kinase (ERK), and phosphoinositide-3-kinase-protein kinase B (PI3K-PKB/Akt) cascades [218]. Among the phlorotannins, dieckol was reported as the most effective in terms of showing anti-neuroinflammatory effects by decerasing pro-inflammatory enzymes by inhibiting the activation of MAPK and NF-κB [146]. In a similar manner, phlorotannins 6,6′-bieckol and phlorofucofuroeckol-B showed significant anti-neuroinflammatory properties, predominantly by downregulating MAPK and NF-κB signaling cascades in combination with a precise reduction in the expression of pro-inflammatory proteins and cytokine generation [217,219].

## 6. Conclusions

Marine sources are an appealing option for the food industry, since a range of functional food ingredients can be derived from these sources. Moreover, people are becoming increasingly aware of the importance of diet and good health; therefore, it is expected that the consumption of functional foods is likely to greatly increase. Marine-derived compounds are an attractive option for the development of functional food ingredients for multiple reasons, including cost-effective extraction processes, their natural occurrence, and their wide availability. Various in vitro and in vivo studies have demonstrated that marine-derived compounds have great potential to exhibit anti-neuroinflammatory responses against neuroinflammatory diseases. Thus, clinical studies are required on using these marine-derived compounds as food ingredients so that their use in treating or even preventing human neuroinflammatory diseases can be established.

## Figures and Tables

**Figure 1 molecules-28-00002-f001:**
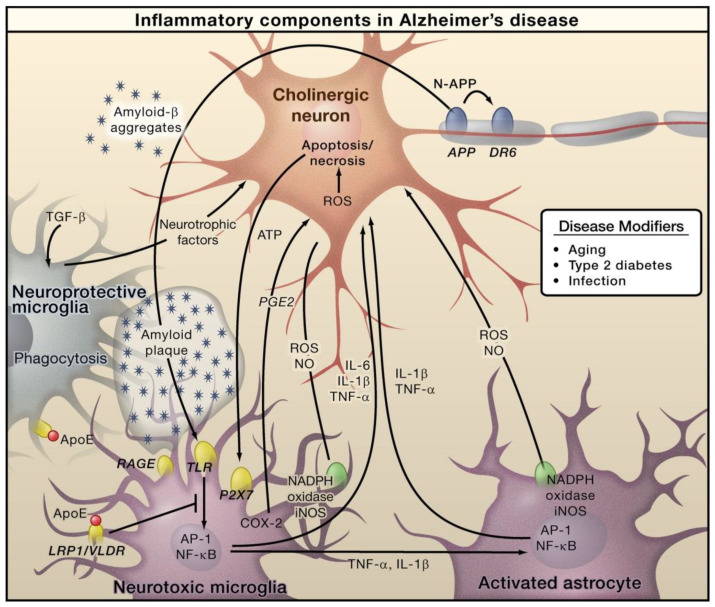
The role of inflammation mediators in the pathogenesis of Alzheimer’s disease. Reproduced with permission from Elsevier [47].

**Figure 2 molecules-28-00002-f002:**
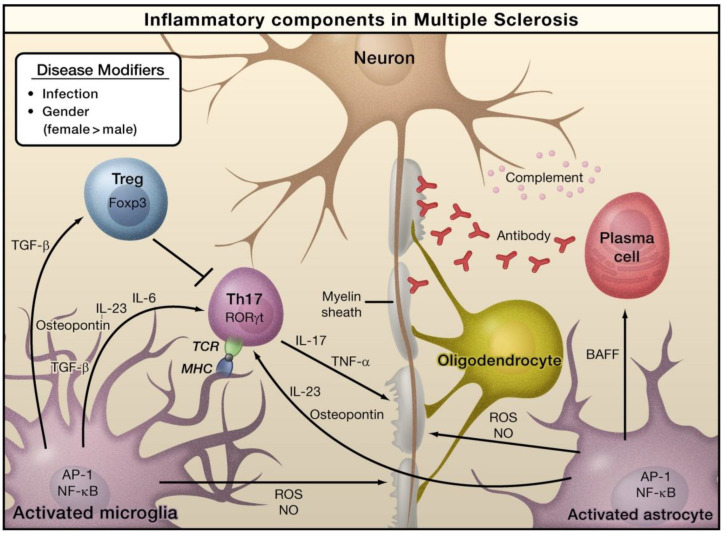
The role of inflammation mediators in the pathogenesis of multiple sclerosis. Reproduced with permission from Elsevier [47].

**Figure 3 molecules-28-00002-f003:**
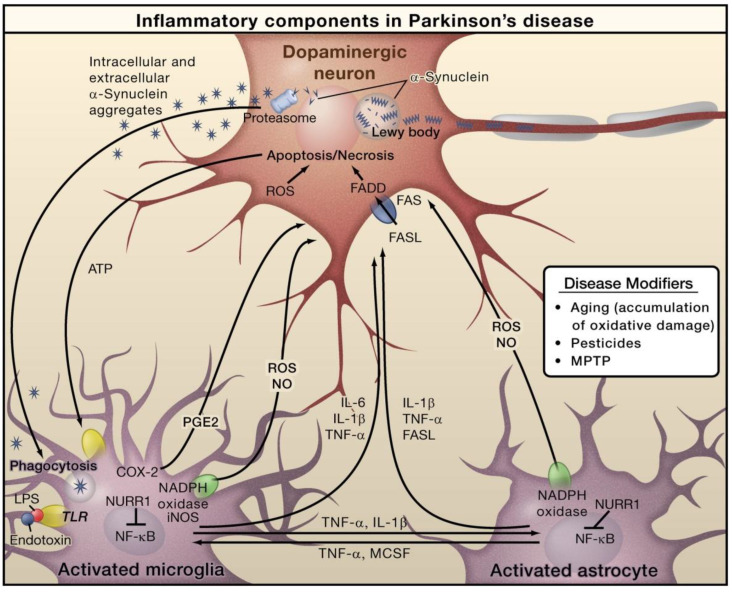
The role of inflammation mediators in the pathogenesis of Parkinson’s disease. Reproduced with permission from Elsevier [47].

**Figure 4 molecules-28-00002-f004:**
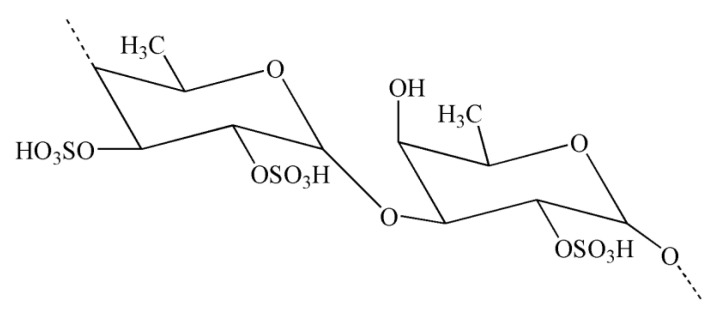
Chemical structure of fucoidan.

**Figure 5 molecules-28-00002-f005:**
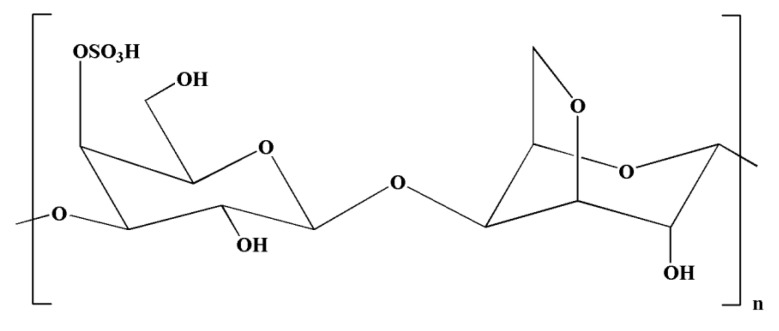
Chemical structure of kappa-carrageenan.

**Figure 6 molecules-28-00002-f006:**
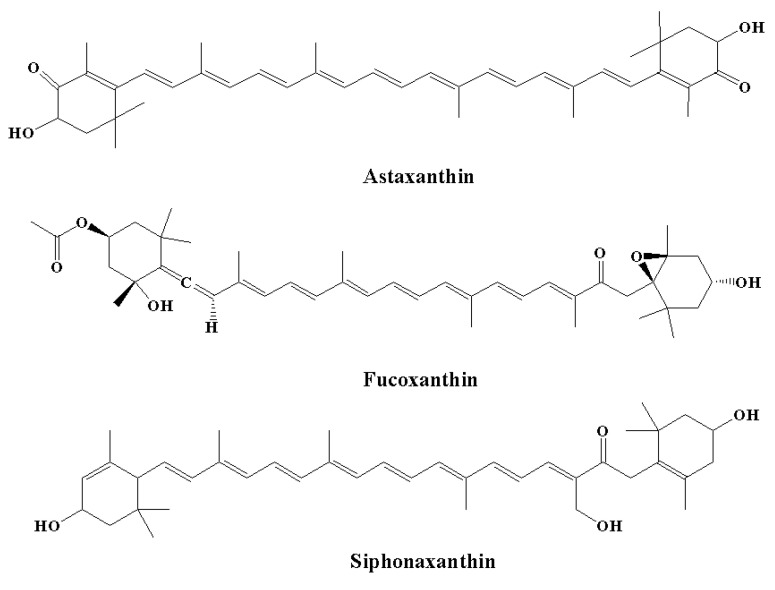
Chemical structures of natural marine pigments with anti-inflammatory potential.

**Figure 7 molecules-28-00002-f007:**
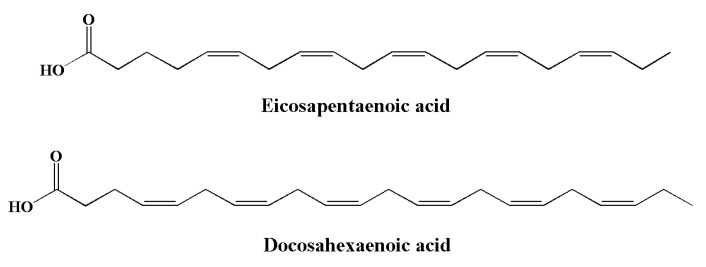
Chemical structures of marine-based omega-3 fatty acids with anti-inflammatory potential.

**Figure 8 molecules-28-00002-f008:**
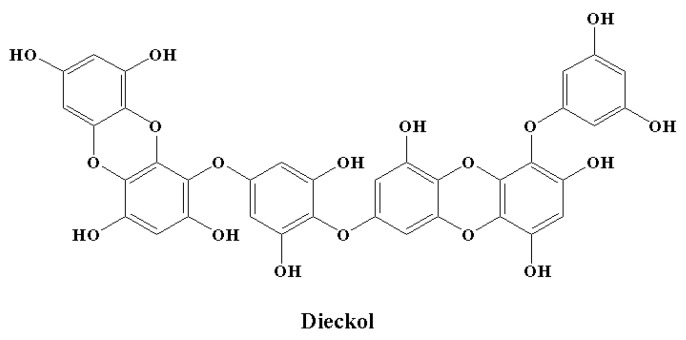
Chemical structure of a marine polyphenol with anti-inflammatory potential.

**Table 1 molecules-28-00002-t001:** Role of inflammatory and immune responses in neuroinflammatory disorders.

Neuroinflammatory Disorders	Inflammatory and Immune Responses	References
Alzheimer’s disease	Elevated levels of TLRs, chemokines, tumor necrosis factor-alpha (TNF-α), and interferon-gamma (IFN-γ); activated and dystrophic microglia	[27,28,29]
Huntington’s disease	Elevated proliferation of microglia; increased concentrations of complement components and interleukin (IL)-6	[30]
Parkinson’s disease	Activation of microglia; increased levels of TNF-*α*, IL-6, IL-1β, cluster of differentiation 14 (CD14), and toll-like receptors (TLRs)	[27,31,32]
Multiple sclerosis	Activation of macrophage and microglia; elevated levels of chemokines, cytokines, and reactive oxygen species (ROS)	[27,33,34,35]
Amyotrophic lateral sclerosis	Increased concentrations of TNF-*α*, IL-6, macrophages, and CD14	[27,34,35]
Stroke	Increased concentration of IL-10	[27]
Schizophrenia	Activation *of* microglia; increased levels of pro-inflammatory cytokines, TLRs, and chemokines	[36,37]
Traumatic brain injury	Increased generation of pro-inflammatory cytokines; elevated levels of inflammasome proteins	[38,39]
Prion diseases	Activation of microglia; generation of various pro-inflammatory mediators, including ROS, NO, IL-1β, IL-6, and TNF-α	[40,41]
Meningitis	Increased levels of TNF-*α* and IL-6	[42]
Epilepsy	Elevated levels of various pro-inflammatory signals, including nuclear factor kappa B (NF-κB) signaling, cell adhesion molecules, toll-like receptors, prostaglandins, chemokines, and cytokines	[12]
Autism	Increased concentration of pro-inflammatory cytokines, including IL-6, TNF-α, IFN-γ, and IL-1β	[43]
Depression	Elevated levels of chemokines and cytokines	[44]
Bipolar disorder	Increased levels of TNF-*α* and pro-inflammatory cytokines; activation of microglia	[45]

**Table 2 molecules-28-00002-t002:** Marine-derived functional food ingredients and their marine sources.

Functional Food Ingredients	Marine Sources	References
Polysaccharides	Seaweeds, microalgae, macroalgae, chordate, cyanobacteria, and invertebrates	[69]
Pigments (carotenoids)	Marine organisms including fungi, algae, archaea, and bacteria	[70,71]
Proteins, peptides, amino acids	Crustaceans, algae, fish frame, marine invertebrates, fish, and algae protein waste	[72,73,74,75,76,77]
Omega-3 fatty acids	Fish, algae, and mussels	[78,79]
Polyphenols	Algae	[80]

**Table 3 molecules-28-00002-t003:** Summary of the potential marine-derived compounds that could be used as functional food ingredients against neuroinflammatory disorders.

Ingredient Family	Individual Ingredient	References
Polysaccharides	Fucoidans, kappa-carrageenan oligosaccharides	[132,133]
Pigments (carotenoids)	Astaxanthin, fucoxanthin, siphonaxanthin	[134,135,136,137]
Proteins, peptides, and amino acids	PPY1, lectins	[138,139,140]
Omega-3 fatty acids	Eicosapentaenoic acid, docosahexaenoic acid	[141,142,143,144,145]
Polyphenols	Dieckol	[146]

## Data Availability

Not applicable.

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
