# Peer review of "Anti-Inflammatory Effects of Marine Bioactive Compounds and Their Potential as Functional Food Ingredients in the Prevention and Treatment of Neuroinflammatory Disorders"

_molecules, 2022, doi:10.3390/molecules28010002_

Round 1

Reviewer 1 Report

Quite relevant review. Marine-derived compounds really are a rich source of bioactive properties.

The review made an interesting compilation on anti-inflammatory activity, focusing on neuroinflammation.

I recommend a professional revision of the English language and standardization of tables according to the standards of the journal

Reviewer 2 Report

Title: Anti-inflammatory effects of marine-derived compounds as functional food ingredients and their roles in the prevention and treatment of neuroinflammatory disorders

In this manuscript, the anti-inflammatory potential of marine-derived compounds as functional food ingredients in the prevention and treatment of neurological disorders has been covered in this review. The idea of this article is good, but the structure level of some articles needs to be revised again. For this reason, I believe that major revisions are required before this manuscript can be considered for publication.

 1.         Figures 1 to 3 are not described in detail in the article.

2.   Functional food ingredients from marine sources, the article level of this part needs to be adjusted. It is recommended to write in four directions: (1) source of functional food ingredients; (2) characteristics; (3) physiological activity; (4) current application.

3.         For the 3.1 polysaccharide part, it is recommended to add a subtitle according to the content.

4.         Line 154-157. “The polysaccharide level in seaweeds can vary from 476% of dry weight dependent on the harvesting time, cultivation method, location, water temperature, season, geographical origin, and macroalgae species.” It is not clear what the author wants to express in this sentence, so it is suggested to revise it.

5.         The functional food ingredients in Table 2 should be consolidated according to the subheadings in Title 3, so it is recommended that Table 2 be revised.

6.         Suggestion 3. Add a summary of functional food ingredients from marine sources, and then put the revised Table 2 in this paragraph.

7.         Some titles have boldface and some don't, should be consistent.

8.         Line 261. 3.3. Proteins, Peptides, and Amino Acids are recommended to be presented under a common name.

9.         Suggestions (1) 4. Add a summary of the potential of marine-derived compounds as functional food ingredients against neuroinflammatory disorders. (2) Organize the content of this part of the article into a table and place it in this paragraph.

10.     The introduction order of the functional ingredients in the third part and the fourth part is inconsistent (such as 3.2. Pigments, 4.2 Polyphenols), and the recommended order of introduction is the same.

11.     Why are there so many functional ingredients and only put the chemical structure formula of specific materials (Figure 4 to Figure 6)? Is there a special reason?

Reviewer 3 Report

Give the latest research on each compound

Give the mechanism of action of each compound for Anti-inflammatory action in  neuroinflammatory disorders 

Reviewer 4 Report

The review MS: Anti-inflammatory effects of marine-derived compounds as 2 functional food ingredients and their roles in the prevention 3 and treatment of neuroinflammatory disorders' is novel and well prepared. However, I have the following comments:

Abstract is too long: remove definition like ''The term neuroinflammation denotes the condition characterized 18 by the increased level of proinflammatory mediators within the central nervous system''. Please revamp and make it more comprehensive 

The word functional food is common, yet it to be confused with ''functional food ingredients'' which has been used in the title. What is the difference. Please reconcile and clarify this in the MS

How the review prepared. Where were the data mining done? How literature was searched. Please provide a paragraph on this

Permission for using figures should be attached as supplementary files

In the introduction, it is important to identify the research gap, why this review is important. It should be clear how this review will advance knowledge 

Round 2

Reviewer 2 Report

Title: Anti-inflammatory effects of marine-derived compounds as functional food ingredients and their roles in the prevention and treatment of neuroinflammatory disorders

After re-reading the manuscript of "molecules-2048968-peer-review-v2", I noticed that the authors made substantial revisions. In general, the article is well-structured and clearly described. After a minor revision this manuscript may be considered for molecules.

1. Line 403-410. The explanation in this paragraph should be placed under "5. Potential of marine-derived compounds as functional food ingredients against neuroinflammatory disorders", as a summary of the fifth part, and Table 3 should be added.

2. Line 413. Table 3 should be deleted.

3. Line 566. 5.3 Bioactive peptides should be revised to Proteins, Peptides, and Amino Acids.
